# Cell and Cell Free Therapies in Osteoarthritis

**DOI:** 10.3390/biomedicines9111726

**Published:** 2021-11-19

**Authors:** Pau Peláez, Elena Damiá, Marta Torres-Torrillas, Deborah Chicharro, Belén Cuervo, Laura Miguel, Ayla del Romero, Jose Maria Carrillo, Joaquín J. Sopena, Mónica Rubio

**Affiliations:** 1Bioregenerative Medicine and Applied Surgery Research Group, Department of Animal Medicine and Surgery, CEU Cardenal Herrera University, CEU Universities, C/Tirant lo Blanc, 7, Alfara del Patriarca, 46115 Valencia, Spain; pau.pelaez@uchceu.es (P.P.); marta.torrestorrillas@uchceu.es (M.T.-T.); debora.chicharro@uchceu.es (D.C.); belen.cuervo@uchceu.es (B.C.); laura.miguel@uchceu.es (L.M.); ayla.delromero@uchceu.es (A.d.R.); jcarrill@uchceu.es (J.M.C.); jsopena@uchceu.es (J.J.S.); mrubio@uchceu.es (M.R.); 2Garcia Cugat Foundation CEU-UCH Chair of Medicine and Regenerative Surgery, 08006 Barcelona, Spain

**Keywords:** osteoarthritis, cell-based therapy, cell-free therapy, platelet-rich plasma, mesenchymal stem cells

## Abstract

Osteoarthritis (OA) is the most common articular disease in adults and has a current prevalence of 12% in the population over 65 years old. This chronic disease causes damage to articular cartilage and synovial joints, causing pain and leading to a negative impact on patients’ function, decreasing quality of life. There are many limitations regarding OA conventional therapies—pharmacological therapy can cause gastrointestinal, renal, and cardiac adverse effects, and some of them could even be a threat to life. On the other hand, surgical options, such as microfracture, have been used for the last 20 years, but hyaline cartilage has a limited regeneration capacity. In recent years, the interest in new therapies, such as cell-based and cell-free therapies, has been considerably increasing. The purpose of this review is to describe and compare bioregenerative therapies’ efficacy for OA, with particular emphasis on the use of mesenchymal stem cells (MSCs) and platelet-rich plasma (PRP). In OA, these therapies might be an alternative and less invasive treatment than surgery, and a more effective option than conventional therapies.

## 1. Introduction

Osteoarthritis (OA) is the most common articular disease in adults, and it affects around 250 million people all over the world, with high prevalence of asymptomatic patients [1,2]. In addition, pain associated with this pathology is one of the main causes of disability worldwide [3]. OA is subcategorized into primary (idiopathic) and secondary. The most common causes of secondary OA are post-traumatic, dysplastic, infectious, and inflammatory processes. Moreover, articular cartilage injuries may lead to the early onset of OA and have a huge negative impact on patients’ function and life quality [4]. Even though the etiology is not completely elucidated, genetic factors and physiological changes such as age, ethnicity, and biomechanical factors play a very important role in its onset [5]. Moreover, the prevalence of OA is higher in women than in men, and the risk of developing OA increases after menopause [6].

In this degenerative disease, the proinflammatory cytokines, such as tumor necrosis factor alpha (TNF-α) and interleukin-1 β (IL-1β), are involved in an imbalance between anabolic and catabolic agents. Moreover, these cytokines stimulate the secretion of enzymes and other cytokines in chondrocytes and synoviocytes, which leads to the spreading of the inflammatory process into the whole joint. The consequence is the destruction of the matrix components, such as proteoglycans and type II collagen [7].

OA pathophysiology includes osteophyte formation, subchondral bone remodeling, synovial inflammation, and the degradation of articular cartilage, culminating in a loss of normal joint function [8]. The articular cartilage has an avascular, aneural and alymphatic nature, and chondrocytes are the only cell type in the extracellular matrix (EMC) [8,9]. After cartilage damage, the repaired tissue has a fibrous nature and does not have the functional characteristics of native hyaline cartilage, leading to articular degeneration [10]. Due to the limited capacity for spontaneous cartilage regeneration, currently, there are no effective treatments for OA management. Pharmaceutical therapies, such as analgesics, nonsteroidal anti-inflammatory drugs, chondroitin sulfate and glucosamine (symptomatic slow-acting drugs), have limited efficacy in halting OA progression; moreover, non-pharmacological therapies (exercise, weight management, education) are underutilized frequently [11]. On the other hand, surgical treatments (arthroscopic debridement, microfracture/osteoplasty, and chondrocyte implantation techniques, such as autologous chondrocyte implantation and matrix-induced autologous chondrocyte implantation) have uncertain effectiveness in cartilage regeneration [12]. Furthermore, recently there have been investigations concerning the chondrogenic characteristics of cartilage progenitor cells [13,14].

Bioregenerative therapies, such as Mesenchymal Stem Cells (MSCs) and platelet-rich plasma (PRP), are a good therapeutic strategy to treat OA [10] (Figure 1)**.** MSCs, a cell-based therapy, improve OA symptoms by decreasing inflammation and pain, and they also stimulate the local repair and regeneration of damaged joint tissues, and secrete bioactive molecules, such as trophic factors, cytokines (including transforming growth factor beta-TGF-β), and pro-angiogenic and antioxidant substances [15]. It has been demonstrated that TGF-β can stimulate the differentiation of MSCs towards chondrocytes that express type II collagen, which provides joint resistance. Furthermore, this regenerative therapy enhances chondrogenesis and the reparation of full-thickness cartilage defects [16,17]. On the other hand, PRP, a cell-free therapy, plays a fundamental role in OA management. The safety and low incidence of adverse effects makes it an appropriate treatment for patients with OA, especially when other therapeutic options are contraindicated [18]. PRP contains growth factors (GFs), cytokines and chemokines that have an anabolic effect on the resident cells, and it can inhibit inflammation, relieving OA symptoms [19]. For example, TGF-β is involved in matrix production, cell proliferation and osteochondrogenic differentiation; Insulin-like growth factor-1 (IGF-1) modulates chondrogenesis and induces MSCs proliferation; Fibroblast Growth Factor-2 (FGF-2) stimulates the proliferation of chondrocytes and MSCs; and platelet-derived growth factor (PDGF) performs its mitogenic function by stimulating cell proliferation and proteoglycan production [20].

MSCs and PRP treatments are commonly used in clinical practice due to their efficacy and safety. The safety of these products is ensured thanks to the Food and Drug Administration (FDA) that regulates these therapies via section 361 of the Public Health Service Act [21]. No FDA approval is required for tissue products and human homologous cells with minimal manipulation, because there is no structural alteration or changes in their biological nature. Nevertheless, formal FDA approval is needed to culture cartilage cells in a laboratory, and for transplantation or allogenic applications [22]. 

Due to the high prevalence of OA and the limitations of conventional treatments in halting the progression of the disease, this review focuses on cell-based therapy and a cell-free therapy as an alternative to medical or surgical conventional treatments, with particular emphasis on the use of MSCs and PRP. 

## 2. Literature Search Methods

The authors conducted research in PubMed, Web of Science and Scopus databases for English-language articles using a combination of the following keywords: “osteoarthritis”, “cell-based therapy”, “platelet-rich plasma” and “mesenchymal stem cells”. A primary selection based on OA and treatments was made, and studies including the use of cell therapy to treat OA were selected. Around 600 articles have been published in the last five years. Special attention has been paid to original analyses and studies, and a total of 184 articles were selected. References of articles found in the primary and secondary search were also included. 

## 3. Bioregenerative Therapies in Osteoarthritis

There is evidence to suggest improvements in osteoarthritic patients after treatment with bioregenerative therapies. Beyond conventional therapies, such as pharmaceutical and surgical procedures, there is an increasing number of alternative treatments, such as the use of MSCs and PRP.

MSCs have been renamed as “medicinal signaling cells” because these cells rest in the perivascular niche until signals are produced after damage. The MSCs migrate to the injury site and produce bioactive molecules, which are involved in reestablishing tissues’ homeostasis [23]. 

Stem cells have been shown to have different mechanisms of action: they have a paracrine activity thanks to the secretion of bioactive molecules such as GFs, cytokines, and pro-angiogenic and antioxidant substances; they stimulate cell proliferation and vascularization in damaged tissues; they promote injury repair because of their capacity to differentiate into other cell lineages; and they regulate inflammatory and immune responses [24,25]. In OA, MSCs play an important role due to their immunoregulatory and regenerative characteristics [26]. They can restore the lost cartilage volume by stimulating the synthesis of angiogenic, mitogenic, antiapoptotic and antiscarring mediators. In addition, it has been suggested that MSCs are involved in Micro-RNA secretion, and they therefore regulate the gene expression of surrounding cells. Moreover, they can control the inflammation process via immune modulation [27,28]. 

PRP is a cell-free therapy used to treat OA. It is an autologous product that contains GFs and several proteins that play an important role during tissue repair and regeneration processes, such as fibrinogen, fibronectin, and vitronectin. These proteins allow the adhesion of cells and other molecules, which are useful for cell conduction, acting as a support “matrix” for tissue repair [29]. When PRP is IA injected, it provides a three-dimensional network in the joint composed of fibrin, which contains binding sites for cell adhesion, as well as proteins, which create a microenvironment that leads to the adhesion of different molecules and cells [19]. Consequently, PRP improves pain and function in OA patients [30,31].

The combination of MSCs with PRP has some beneficial effects, such as increased collagen type II expression and decreased chondrocyte apoptosis. Furthermore, PRP has an anabolic effect on both MSCs and chondrocytes, assisting in proliferation, inhibiting deregulation, and supporting matrix development, which further maintains appropriate chondrocyte and stem cell development [32]. This combination leads to a reduction in TNF-α and IL-6 concentration, together with an increase in proteoglycans concentration in articular cartilage [33]. 

### 3.1. Mesenchymal Stem Cells in Osteoarthritis

The use of MSCs to treat OA has been demonstrated to be safe, and its use has increased in the last years compared to conservative treatments [34]. MSCs secrete trophic factors with regenerative functions, and they can differentiate into cartilage and bone cells, promoting cartilage regeneration [35].

In animal studies, the capacity of MSCs to stimulate local repair and cartilage regeneration in damaged joints has been demonstrated [36,37]. Moreover, regarding human medicine, Lendeckel et al. described the use of adipose-derived stem cells (ADSCs) combined with bone graft and fibrin glue to treat cranial critical-size bone defects in a seven-year-old girl suffering from multiple calvarial fractures. Three-month follow-up computed tomography scans revealed new bone formation and almost complete calvarial continuity [38].

In OA, MSCs therapy reduces pain and inflammation [39]. MSCs modulate the inflammatory response via the suppression of inflammatory T-cell proliferation and the inhibition of monocyte and myeloid dendritic cell maturation. Moreover, the presence of a pro-inflammatory milieu has been suggested as the key to promoting MSCs’ anti-inflammatory effects [15,40]. After the IA injection of MSCs, molecules with anti-inflammatory and chondrogenic properties are expressed. These cells could help to establish a regenerative microenvironment at the site of release, which would improve the recruitment, activation, and differentiation of endogenous stem cells with potential to repair the articular cartilage [17].

Although many studies have focused on OA treatment with autologous MSCs, the safety of allogenic MSCs for treating knee OA has also been demonstrated [41,42,43,44].

There are two different types of MSCs, embryonic and adult stem cells. Adults MSCs can be obtained from different tissues such as peripheral blood, bone marrow and adipose tissue, the last two being the most commonly used sources. MSCs found in adipose tissue are called ADSCs, which have a greater proliferation capacity than the rest of the MSCs, and they are able to maintain the differentiation potential for longer periods when cultured. Additionally, their ability to proliferate is not affected by donors’ age as much as in other kinds of MSCs [45], even though it has been shown that aging negatively impacts ADSCs function [46]. Moreover, ADSCs have several advantages over other kinds of MSCs: adipose tissue is abundant and easy to obtain, in vitro culture is not necessary and therefore they have a low risk of rejection [47], and it is an economical product. It has been demonstrated that the IA application of autologous ADSCs with no previous culture halted the progression of cartilage degeneration [48]. Moreover, adipose tissue contains approximately 10 times more MSCs than bone marrow, making this source of MSCs a good candidate for the treatment of OA [15]. In a recent study, the efficacy and safety of Bone Marrow Mesenchymal Stem Cells (BMMSCs) and ADSCs have been compared. Data from 19 studies including a total of 811 patients with knee OA were analyzed. At 6-month follow up, ADSCs showed significantly greater improvements in visual analog scale (VAS) and Western Ontario and McMaster Universities Osteoarthritis Index (WOMAC) than BMMSCs, compared to controls. At 12-month follow up, ADSCs outperformed BMMSCs compared to their controls in measures such as WOMAC, knee injury and osteoarthritis (KOOS), and Whole-Organ Magnetic Resonance Imaging Scores (WORMS). Similar results were observed at 24-month follow up, where ADSCs showed significantly better Lysholm scores than BMMSCs, although VAS improvement was better with BMMSCs [49].

The stromal vascular fraction (SVF) derived from adipose tissue might induce IA fatty tissue homeostasis. Moreover, it has protective and anti-inflammatory functions, inhibiting OA progression [50]. The SVF contains 500,000 to 2,000,000 cells per gram, including macrophages, monocytes, pericytes, fibroblasts and MSCs, of which approximately 1–10% are ADSCs [51,52]. The SVF and pure ADSCs have similar properties, such as anti-inflammatory, angiogenic and immunomodulatory effects, but the heterogeneity of SVF contributes to better therapeutic results [53]. Some studies have pointed out that SVF enhances new cartilage matrix formation [54] and subchondral bone regeneration better that ADSCs [55]. Furthermore, SVF is obtained more easily than pure MSCs, as it does not need special culture conditions to expand [56], which could modify the migration ability of the cells towards damaged tissue [57].

Several studies have reported the benefits of IA injections of MSCs (Table 1 and Table 2) in patients with OA, based on improvements in pain and function, but further studies focused on damaged tissue regeneration are needed to find the proper treatment for arthritic diseases [58].

#### 3.1.1. Adipose-Derived Mesenchymal Stem Cells

ADSCs can be collected in large quantities via lipoaspirates or adipose tissue biopsy from subcutaneous tissue; in addition, infrapatellar fat pads have been considered as a good source of stem cells for cartilage regeneration due to their increased chondrogenic capacity [59,60]. After collection, these cells are isolated and expanded in vitro [61]. Moreover, Spasovski et al. have demonstrated that the use of ADSCs from subcutaneous fat in patients with knee OA improves clinical symptoms and reduces pain after 3 months, the best results being obtained at 6-month follow up [62]. Chondrogenesis potential has been reported for both infrapatellar and suprapatellar ADSCs [63,64], although infrapatellar ADSCs showed better results both in vitro and in vivo [64,65]. In addition, the suprapatellar-derived ADSCs transplantation in a mouse model of severe knee OA decreased inflammation and cartilage degenerative grade, increased the synthesis of glycosaminoglycans, and induced endogenous chondrogenesis [63]. These effects may be a consequence of the reduction in pro-inflammatory cytokines and chemokines, and the apoptosis of chondrocytes, hypertrophic and fibrotic chondrocyte phenotypes and collagenases mediated by ADSCs [66].

Several studies have demonstrated the effectiveness of MSCs in OA treatment. In relation to ADSCs, the first case report was published in 2001 [67]. Since then, these cells have attracted great interest because they have been demonstrated to be safe and efficient for articular cartilage regeneration. In this sense, some reports showed that the IA injection of ADSCs promotes cartilage regeneration, thus, a single IA injection of ADSCs in 9 patients with knee OA exhibited improvements in clinical scores 6 months after treatment [62]. Lee et al. published a study wherein ADSCs were administered in 12 patients (MSCs group), and the MSCs group was compared with a control group (12 knees that were injected with saline solution); at 6-month follow-up, a single IA injection of ADSCs was associated with improvement in WOMAC score as compared to baseline. Patients in the control group did not show significant improvements at 6 months [68].

Takagi et al. showed in an anterior cruciate ligament transection model in rabbits that IA ADSCs sheets halted OA progression. To make ADSCs sheets, a medium containing ascorbate-2-phosphate was used to enhance collagen protein secretion. Four weeks after anterior cruciate ligament transection, autologous ADSCs sheets were IA injected into the right knee (ADSCs sheets group), and autologous cell death sheets treated by liquid nitrogen were injected into the left knee (control group). Injections were administered once a week. Rabbits were sacrificed 6, 8, 10 and 12 weeks after an anterior cruciate ligament transection (6 rabbits per week), and femoral condyles from both knees were harvested and macroscopically and histologically compared. Macroscopically, OA progression was significantly lower in the ADSCs sheets group than in the control group, and histologically, control knees showed obvious erosions in the medial and lateral condyles, while cartilage integrity was maintained in the ADSCs sheets group [69].

Moreover, a positive correlation between ADSCs dosage and clinical improvement has been demonstrated. In a study comparing three different autologous expanded ADSCs doses (10 × 10^6^, 50 × 10^6^, 100 × 10^6^) in patients with knee OA, the higher-dose group showed significantly greater improvements in terms of pain, function and mobility compared with lower-dose groups [70]. Accordingly, 18 patients treated with three different ADSCs doses (1 × 10^7^, 2 × 10^7^ and 5 × 10^7^) showed the greatest improvements in knee joint cartilage volume, pain, and function with the highest dose [71]. Furthermore, another study that consisted of three consecutive cohorts (six patients with knee OA) with dose escalation (low dose (2 × 10^6^ cells), medium dose (10 × 10^6^), and high dose (50 × 10^6^)) showed that the group of patients injected with the low dose exhibited the best response to ADSCs treatment, whereas they had higher baseline pain and WOMAC scores compared with those receiving higher doses. One possible reason for this inverse dose effect might be the higher level of inflammation in the lowest dose group, as reflected by the highest level of pain at baseline. The inflammatory milieu might have primed the injected ADSCs to exert their immunomodulatory functions more efficiently than in the groups where the inflammation was lower [72]. In the same line, in a recent study conducted in thirty patients with knee OA separated into three groups (*n* = 10) (the control group received conventional conservative management, the first treatment group received one ADSCs injection (1 × 10^8^) and the second treatment group received two injections 6 months apart (1 × 10^8^ ADSCs)), both treatment groups receiving ADSCs showed clinically significant improvements in pain and function at 12 months follow-up, and radiological analysis using the MRI Osteoarthritis Knee Score (MOAKS) indicated modification of disease progression [73].

On the other hand, IA injection of allogenic ADSCs combined with HA could stop OA progression and promote cartilage regeneration [74]. The use of allogenic ADSCs in a rabbit model has demonstrated that these cells are able to adhere to the osteochondral defect and promote histological healing [75]. Moreover, allogenic ADSCs survival rates of 14 weeks after IA injection were reported in a sheep model, which leads to cartilage regeneration [74]. Recently, in 58 dogs, visceral adipose tissue (a surgical waste obtained during routinary ovariectomy) served as a source of allogeneic ADSCs and was used to treat OA. In total, 83% of the dogs showed improvements in lameness at 4–5-years follow-up [76]. Finally, another study in dogs with elbow dysplasia treated with allogeneic ADSCs showed significant improvements in range of motion 42 days after infiltration; furthermore, the analysis of the concentration of several biomolecules, such as metalloproteinase-3 (MMP-3), IL-6 and TNF-α, in the synovial fluid showed significant differences before and after the treatment [77].

The combination of arthroscopic abrasion arthroplasty with IA injection of ADSCs (50 × 10^6^ at baseline and at 6 months) in 27 patients with advanced knee OA showed significant clinical improvements in pain and function, and hyaline-like cartilage regeneration was observed in all participants by MRI [78]. Moreover, IA injection of ADSCs combined with arthroscopic debridement in patients with early knee OA has reported good results in terms of pain and function [79]. In addition, a combination of ADSCs with PRP in a rat model has shown that PRP potentiates the effect of ADSCs in damaged articular cartilage [33]. Recently, Okamoto-Okubo et al. reported the best results with allogenic ADSCs over PRP in dogs with bilateral degenerative hip joint disease [80].

ADSCs have shown excellent results in the treatment of chondral defects, and the successful management of a post-traumatic chondral defect using IA autologous ADSCs therapy has been suggested [81]. Moreover, Freitag et al. obtained promising results in a patient with knee osteochondral defect due to osteochondritis dissecans, who showed improved osteochondral architecture and cartilage volume after MRI after treatment [82].

One of the main limitations of the studies that describe the use of ADSCs in OA is the short follow-up period. Pers et al. reported the efficacy of IA injections of these cells for knee OA treatment, but with a follow-up period of only 24 weeks [72]. Recently, Song et al. published the first study with a long-term follow-up (96 weeks), and it has demonstrated the efficacy of ADSCs therapy in knee OA with repeated IA injections. These patients showed improvements in pain, function, and cartilage volume [71]. In other studies with 4–5-year follow-ups, improvements in function have been reported [76].

In the last few years, the efficacy of intravenously administered ADSCs in dogs with elbow OA has been studied, with no significant improvements in synovial fluid biomarker concentrations and mean peak vertical force, so further investigation is needed in this field [83]. Recently, a scaffold-free 3D-printed construct consisting of autologous ADSCs has shown the ability to regenerate cartilage and subchondral bone within 3 months in an osteochondral defect surgically induced in the left femoral trochlear groove of rabbits [84].

ADSCs in OA may offer an interesting possibility to improve function, pain, and cartilage volume of the joint, suggesting that they are a good therapeutic strategy for OA.

#### 3.1.2. Stromal Vascular Fraction

The SVF is a product of heterogeneous composition, and the most abundant cells are ADSCs, blood cells, fibroblasts, macrophages, pericytes, endothelial cells and their progenitors. Each cell type has a specific mechanism of action, which is complementary to the others. This cellular heterogeneity gives SVF a high therapeutic potential. In addition, the high concentration of MSCs present in the SVF allows its use for therapeutical purposes without needing previous culturing, which makes it an attractive product for clinical practice [12].

In osteoarthritic patients, SVF has an anti-inflammatory effect on synoviocytes and chondrocytes due to the release of anti-inflammatory molecules such as interleukin-1 receptor antagonist (IL-1Ra), indoleamine-2, 3-dioxygenase, TGF-β and prostaglandin E2 (PGE2) [85]. SVF has been proposed as a promising candidate for halting the progression of degenerative joint disease, because the adipose tissue of the infrapatellar fat pad interacts with the SVF, inducing a cascade of structural and molecular responses that lead to an anti-inflammatory a homeostatic environment [50]. In this sense, Pak et al. showed functional and clinical improvements after IA injection of SVF in three patients with knee OA with a 2-year follow-up period [86]. In another study, six patients with knee OA showed functional improvements 3 months after SVF administration, and a decrease in pain after 1 year follow-up [87]. Another clinical study with 57 patients with knee OA treated with IA injection of 2.5 × 10^7^ SVF showed improvements in WOMAC, VAS and KOOS score at 1, 3, 6 and 12 months after injection [88]. Moreover, adipose tissue-derived SVF has been proposed as a novel therapeutic option in patients with OA in the temporomandibular joint. To evaluate the anti-inflammatory effect of high-quality SVF, an in vitro study was performed to assess the expression patterns of inflammatory cytokines, including PGE2, interleukin-6 (IL-6), and C-X-C Motif Chemokine Ligand 8/interleukin-8 (CXCL8/IL-8), in a co-culture with synoviocytes derived from the synovial fluid in the temporomandibular joints of patients with OA, and a significant down-regulation of certain inflammatory cytokines such as PGE2 and CXCL8/IL-8 in the temporomandibular joint was reported [89].

In a study with 1128 patients and a total of 1856 knee and hip osteoarthritic joints, favorable clinical results were assessed 3 to 12 months after SVF infiltration, using a prefabricated system for SVF obtention, although neither the anatomical nor histological effects of SVF infiltration on cartilage were evaluated [53]. In addition, it should be considered that the limited number of patients in most studies, together with the combination of SVF infiltration with other treatments, hampers the evaluation of SVF’s effect on these patients. It has been demonstrated that 1 year after IA injection of SVF and PRP in 10 patients with knee OA, 6 of them showed an increase in cartilage thickness, while all participants had functional and pain improvements at 2 years of follow-up [90]. Additionally, 16 patients with bilateral knee OA were included in a study wherein one knee was treated with SVF, and the contralateral joint was treated with HA. The study confirmed that SVF improves function, relieves pain, and repairs cartilage defect compared with HA-treated knees [91]. The combination of a microfracture technique with SVF infiltration has shown better results than microfracture alone in 33 patients with knee OA. A decreasing trend in VAS score and WOMAC index in the SVF-treated patients was observed 24 months after treatment [92]. Furthermore, the IA application of autologous-microfragmented adipose tissue (AMFAT) with SVF has been investigated in osteoarthritic knees of 17 patients (32 treated knees). Improvements in cartilage quality assessed by delayed gadolinium-enhanced MRI (dGEMRIC) protocol, which determines the glycosaminoglycan content in cartilage before and after treatment, were reported, as well as decreased VAS scores at 12 months follow-up [93]. Recently, another similar study was conducted. Functional scores were measured at baseline and 12 months after treatment, and the results showed improvements in KOOS score, WOMAC index and VAS rating in 85% of the patients, while the remaining 15% of patients underwent a total knee replacement [94]. In this line, Boric et al. published a study with 10 patients and 18 treated osteoarthritic knees, and 24 months after injection with AMFAT, the dGEMRIC and VAS index measurements suggested improvements in function and pain [95].

The SVF regenerative activity may be due to ADSCs’ differentiation potential in osteocytes and chondrocytes. Furthermore, SVF cells lead to tissue remodeling because they have immuno-modulator and anti-inflammatory properties [96].

#### 3.1.3. Bone Marrow Mesenchymal Stem Cells

Bone marrow aspirate (BMA) is one of the most widely used sources in cell therapy because of its easy accessibility. It is usually obtained by aspiration of the iliac crest [97]. There are different alternatives for obtaining a higher number of stem cells from a BMA sample—expanding them in in vitro culture medium to obtain BMMSCs, or processing techniques, such as density separation. Currently, autologous BMA concentrate (BMAC) preparations are made for direct intraoperative use to implant BMA with minimal manipulation [98]. Moreover, BMAC can be easily extracted by FDA-approved commercialized kits [99]. Nonetheless, a drawback of BMAC is the heterogeneous cell population found in the preparation, including endothelial, hematopoietic, and inflammatory cells. In addition, the preparations also vary between individuals, and depend on age, sex, and aspiration site within the same patient [100]. For this reason, we focused our review on BMMSCs.

BMMSCs have many characteristics that make them an optimal product for regenerative therapy, which have been wildly described, including their multipotency, and their anti-inflammatory and immune-modulatory properties [101]. These stem cells have been widely used to treat chondral defects [102]. Several studies in OA patients treated with BMMSCs showed promising results, such as decreases in synovial inflammation and pain, increasing patient satisfaction. Doyle et al. carried out a review to evaluate the efficacy of IA injections of these cells to treat knee OA, and concluded that low confidence in clinical efficacy is due to the heterogeneity of used methodologies, but most studies demonstrated a decrease in poor cartilage index, and improvements in pain, function, and quality of life, with moderate to high levels of evidence regarding safety for therapeutic administration [103]. Recently, a meta-analysis that included 724 patients reported improvements in pain level as measured by VAS, International Knee Documentation Committee (IKDC) function score, Tegner Activity Scale (TAS) and Lysholm Knee score, when the results were compared with the ones obtained before treatment with BMMSCs [104]. On the other hand, Zhi et al. demonstrated that articular cartilage chondrocytes cocultured with BMMSCs could repair cartilage lesions and prevent the abnormal expression of KDM6A and SOX9 (key factors in the pathogenesis of OA), type II collagen, and aggrecan in rats with OA [105].

Regarding the regulation of the inflammatory process in OA, Zhang et al. reported that the co-cultivation of BMMSCs with chondrocytes from patients with OA increases chondrocytes’ proliferation and inhibits inflammatory activity in OA [106]. Moreover, a study with 13 patients suffering knee OA treated with in vitro-expanded BMMSCs showed a significant improvement in femoral and tibial plates’ cartilage thickness at 12-month follow-up [107]. Furthermore, Soler et al. have shown that IA injection with ex vivo-expanded autologous BMMSCs in patients with knee OA improved pain and range of motion [108]. Good results have also been reported with the combination of BMMSCs and microfracture in large [109] and small joints [110].

Some studies assessed different doses of BMMSCs. Clinical trials have demonstrated that high doses (25 × 10^6^ cells and 40 × 10^6^ cells) of autologous BMMSCs are safe [104,111,112] Lamo-Espinosa et al. studied two BMMSCs doses in patients with knee OA, 1 × 10^7^ and 1 × 10^8^, and demonstrated no clinical differences between groups after 4-year follow-up [113]. The same author tried different doses of autologous expanded BMMSCs, 10 × 10^6^ and 100 × 10^6^, in association with HA (control group) in 30 patients with knee OA, and improvements in WOMAC were seen only in patients treated with the highest dose, while improvements in VAS scores were reported in both groups at 12-month follow-up [114]. Another study has demonstrated improved knee function, pain, and quality of life at the 24-month follow-up after two IA injections (with a 1-month interval) of 30 × 10^6^ BMMSCs in patients with knee OA [107]. Different doses of allogenic BMMSCs (25, 50, 75 and 150 × 10^6^ cells) followed by IA injection of HA were assessed. No significant differences with a placebo group were reported, but the best improvements were seen in the 25 × 10^6^ cells dose group [112]. Other study found better outcomes in patients with bilateral middle or advanced knee OA after treatment with 8–9 × 10^6^ autologous BMMSCs at 5-year follow-up [109].

With regards to the combination of BMMSCs with other therapies, in a recent study, 60 patients (30 patients per group) diagnosed with knee OA were randomly assigned to three weekly injections of Plasma Rich in Growth Factors (PRGF) (control group), and the BM-MSCs group was formed of patients who received a single IA injection of 100 × 10^6^ autologous cultured BMMSCs in 3 mL Ringer’s lactate solution, followed by an IA injection of PRGF. A weekly single IA injection of PRGF was applied for 2 additional weeks. A follow-up period of 12 months was established, and pain and function were assessed using VAS and WOMAC scores and by measuring knee range of motion. The study concluded that although no statistical differences were observed between groups, only patients treated with BMMSCs in combination with PRGF could be considered as OA treatment responders following Osteoarthritis Research Society International (OARSI) criteria [115]. Moreover, a recent network meta-analysis that evaluated the clinical efficacy of IA injections of HA, leukocyte-poor platelet-rich plasma (LP-PRP), leukocyte-rich platelet-rich plasma (LR-PRP), BMMSCs, ADSCs, and saline (placebo) during 6 and 12 months of follow-up concluded that at 6-month follow-up, ADSCs is the best therapy in terms of pain relief, while LP-PRP was the most effective in terms of functional improvement. At 12-month follow-up, both ADSCs and LP-PRP showed great potential in pain relief; functional improvements were achieved with LP-PRP, and the clinical efficacy of HA was lower than that of the other biological agents [116].

Regarding parenteral administration, a single dose of BMMSCs (1 × 10^6^ per kg) was injected intravenously into 13 patients suffering from refractory rheumatoid arthritis. At 12-month follow up, sufficient immunoregulatory effect of autologous BMMSCs on T cells was reported [117].

It seems that BMMSCs have a chondroprotective influence on OA, which makes them a potential therapeutic modality for OA therapy.

#### 3.1.4. Other Sources of Mesenchymal Stem Cells

Adult MSCs can be obtained from different tissues, such as peripheral blood, bone marrow and adipose tissue, the latter two being the most commonly used sources.

Embryonic MSCs are obtained from human umbilical cord. These cells, known as human umbilical cord-derived MSCs (hUC-MSCs), enhance the proliferation of OA chondrocytes and downregulate the expression of inflammatory cytokines. As a result, the co-culture of hUC-MSCs and OA chondrocytes could be a good therapeutic option for OA treatment [118]. Sadlik et al. suggested that hUC-MSCs could be used as seed cells to induce cartilage regeneration in patients with OA. In this study, collagen scaffolds containing hUC-MSCs were injected via arthroscopic implantation in cartilage defects. MRI images at 1, 5, 6 and 12 months after implantation showed a large amount of regenerated tissue and a fusion between the surrounding cartilage and the subchondral bone [119]. In addition, Park et al. reported that an hUC-MSC-based product was safe and effective for hyaline-like cartilage regeneration in knee OA after 7-year follow-up [120]. Moreover, a different study compared single and multiple IA injections with hUC-MSCs as a treatment for OA, but no significant differences were obtained in the diagnostic imaging assessment; nevertheless, patients’ pain and functionality significantly improved [121].

Regarding animal studies, hUC-MSCs (5 × 10^6^ cells) and HA hydrogel composite were used to treat knee cartilage injury in miniature pigs. Cartilage lesions in the treatment group showed significantly better results, compared to the control group [122]. On the other hand, it has been demonstrated that a single injection of amniotic suspension allografts (ASAs) from allogenic donors in knees suffering from OA leads to better functionality and decreases pain at 12-month follow-up [123].

Synovium-derived mesenchymal stem cells (S-MSCs) have attracted considerable attention due to their high chondrogenic potential, and are recognized as a viable option to repair cartilage [124,125]. Zare et al. conducted a research study on rats wherein animals treated with S-MSCs showed significantly better results in every pathological evaluated criterion compared to control, HA and ADSCs groups [126]. Furthermore, another research study reported the efficacy of S-MSCs in OA treatment, showing a significant reduction in serum inflammatory factors and cartilage protection thanks to the upregulation of chondrogenic gene expression in meniscus chondrocytes [127]. Moreover, Neybecker et al. demonstrated that S-MSCs in collagen sponges have a good capacity to stimulate the induction of chondrogenic genes and ECM synthesis [128]. On the other hand, when comparing the characteristics of S-MSCs from the hip joints of patients with femoroacetabular impingement syndrome (FAIS) and OA, S-MSCs from patients with OA had greater colony numbers but less viability and proliferative potential. They also showed greater osteogenic and adipogenic potentials, whereas those from patients with FAIS showed greater chondrogenic potential [129].

Peripheral blood MSCs (PBMSCs) are a heterogeneous population harvested from peripheral fresh blood. Comparative studies have shown different growth patterns and cell markers when culturing human PBMSCs under different oxygen tension conditions. Under hypoxic conditions, like those found in bone marrow niches, cells express more than 90% of MSCs markers and maintain their potential of trilinear differentiation for chondrogenic, osteogenic and adipogenic tissues. This phenomenon reflects their potential use for cartilage repair, which is hypoxic in nature [130,131]. Moreover, a recent review assessed the use of PBMSCs in cartilage repair and regeneration in vivo, and showed that although all reviewed articles reported that PBMSCs enhances cartilage repair and regeneration, we should maintain a cautious attitude towards the positive therapeutic effects of PBMSCs, considering the deficiency of studies with a high level of evidence, the incomplete assessment system of outcomes, and the combined use of multiple other treatments [132]; so more clinical studies are necessary [118,119]. That said, PBMSCs could be considered a good option in the treatment of OA.

### 3.2. Mesenchymal Stem Cells Exosomes in Osteoarthritis

Exosomes are extracellular vesicles (EVs) surrounded by a phospholipid membrane, either a cell membrane- or an endosomal-derived membrane, that contains different cell-specific receptors, which are important in cell-to-cell communication [133]. MSCs-derived exosomes isolated from different tissue sources are free of cells and have shown therapeutic potential to treat many diseases [134,135]. Pre-clinical in vivo studies have demonstrated positive effects in joints and have confirmed the effectiveness of EVs injections as a minimally invasive therapy [136]. Particularly, MSCs-derived exosomes have cartilage repair properties and can delay OA progression through a variety of mechanisms; for example, stimulating ECM secretion, promoting chondrocyte proliferation, inhibiting chondrocyte apoptosis, and maintaining chondrocyte homeostasis [137]. Recently, it has been demonstrated that IA injection of BMSCs-derived exosomes reduces inflammation and cartilage damage, and inhibits OA progression. These effects are achieved thanks to the phenotypic transformation of macrophages from M1 to M2, together with a decrease in inflammatory cytokines and the release of anti-inflammatory cytokines [138]. A research study carried out by Wang et al. reported that embryonic MSCs-derived exosomes stimulate the synthesis of type II collagen and decrease the production and expression of MMPs with thrombospondin-like motifs-5 (ADAMTS-5), providing a stable ECM in a surgically induced OA model in mice [139]. On the other hand, Tofiño-Vian et al. revealed that EVs isolated from human ADSCs promote chondroprotective functions due to a decrease in MMP activity, decrease the secretion of inflammatory mediators and stimulate anti-inflammatory cytokine IL-10 production [140]. Furthermore, Cosenza et al. reported that both exosomes and BMMSCs microparticles decrease the expression of catabolic and inflammatory markers [141]. Moreover, in a recent study in which MSCs-derived exosomes were used in cartilage repair, an increased matrix deposition and cellular proliferation and better histological scores were observed in animals treated with MSCs-derived exosomes [142].

In an in vitro study carried out by He et al. in rats with OA, significant reconstituted collagen type II and impaired MMP-13 protein expression in knee joint cartilage were observed after exosome treatment, demonstrating a significant chondrocyte proliferation and migration capacity [143]. Additionally, human articular chondrocytes and fibroblast-like synoviocytes isolated from the same OA patients were cocultured in 2D as well as in 3D conditions with fluorescently labeled ADSC-EVs, and analyzed by flow cytometry or confocal microscopy. In both 2D and 3D conditions, fibroblast-like synoviocytes were more efficient in interacting with ADSC-EVs, and 3D imaging showed a faster uptake process. The removal of the HA component from the ECM of both cell types reduced their interaction with ADSC-EVs only in the 2D system, showing that 2D and 3D conditions can yield different outcomes when investigating events wherein ECM plays a key role. These results indicate that studying EVs’ binding and uptake both in 2D and 3D guarantees a more precise and complementary characterization of the molecular mechanisms involved in the process [144].

MSCs-derived exosomes are a promising new cell-free approach to treat OA and other joint conditions. However, further investigations in humans are required to assess the effectiveness and feasibility of this therapy.

Stem cell therapy could become a first-line therapy to treat OA. The efficacy of MSC-based therapies has been related to the paracrine secretion of trophic factors, and exosomes that play a fundamental role in tissue repair.

### 3.3. Platelet-Rich Plasma in Osteoarthritis

PRP is a cell-free and autologous product of fractionated plasma derived from the patient’s own blood, obtained after a specific centrifugation process. The platelets contain alpha granules that are rich in several GFs with chemoattractant and mitogenic functions, which help to attract surrounding cells to injured areas. This product can also contain a high number of leukocytes, which may have a negative effect on tissue regeneration. In fact, there is a wide variation in the reported protocols for the standardization and preparation of PRP, with variable reported efficacy. PRGF could be an alternative approach; it is a 100% autologous and biocompatible preparation, with a moderate platelet concentration and no leukocytes or erythrocytes, and it is elaborated by a single centrifugation process [145,146]. It is considered a safe product due to its autologous origin [134].

Some studies have shown that PRP regulates cartilage repair by stimulating the synthesis of proteoglycans, aggrecans or type II collagen by chondrocytes; the proliferation of synoviocytes; and the chondrogenic differentiation of MSCs. Additionally, it decreases the catabolic effects of cytokines, such as IL-1, or proteolytic enzymes, such as matrix metalloproteinases [18,19,20,29,147,148].

Despite the variability in PRP preparations, several published research and clinical studies have shown excellent outcomes (Table 1 and Table 2). Cuervo et al. showed significant pain relief and an improvement in limb function after a single IA PRGF infiltration in canine patients with OA secondary to hip dysplasia, and this effect was maintained for more than 6 months [148]. Other studies have addressed a maximum effect 180 days after a single IA PRP injection in patients with knee OA [149]. Furthermore, in a study where patients were treated with PRP combined with HA, a significantly greater pain relief was observed in patients receiving the combination of both treatments compared with those only treated with PRP, suggesting that the association of PRP and HA could be a better long-term option [150]. Huang et al. evaluated PRP IA injections once, twice, or three times per month, and demonstrated the positive effects of PRP up to a 12-month follow-up in patients receiving one or two IA injections, whereas these effects were maintained in those patients receiving three IA injections [151]. With regard to variability in PRP preparations, a systematic review to investigate the superiority of PRP over HA concluded that it is necessary to standardize the centrifugation protocol, the activating agents, the administration frequency, and the injected total volume [152].

Scientific evidence exists regarding PRP injections in all stages of knee OA [30,153]. In a study with 214 patients with knee OA, 155 patients were classified as having Kellgren–Lawrence (KL) grade 1 OA, while 59 were grade 2 OA. Patients were treated with three PRP IA injections in knee joints with 4-weeks intervals, and the WOMAC was evaluated at the time of induction and with 6-month intervals. The mean WOMAC score before treatment was 83.05, and after 6 months, it was significantly reduced to 38.84 [154]. Patients with knee OA graded stage 1 or 2 on the KL scale showed improved WOMAC scores at 1, 2, and 6 months after PRP IA injection [155]. A significant pain relief and functional improvement was reported 3 months after PRP treatment, mainly in lower OA grades [151] compared to control groups [156]. Additionally, Cook and Smith compared PRP with traditional knee OA treatments. Patients reported good pain relief for at least 6 months, and commonly for a year or longer following the three PRP injection regimens [157]. In addition, the effect of IA PRP injection was evaluated with VAS pain scores in patients with knee OA by Taniguchi et al., and the results showed that the average VAS pain scores improved at 6-month follow-up from 71.6 to 18.5 (*p* < 0.05), with 80% of patients experiencing a 50% decrease, or higher, in VAS pain scores [158]. Similar results were obtained in a different study wherein a reduction in pain and lameness was observed in human patients treated with PRP with knee OA after 1 year follow up [159]. MRI image studies were carried out by Ornetti et al., showing no OA progression in 73% of the patients treated with PRP [160].

This therapeutic option has also reported beneficial effects in animals. In a recent study, the effects of a single PRP IA injection were evaluated in dogs with knee OA secondary to cranial cruciate ligament rupture. Improvements in kinetics were seen in dogs 3 months after treatment [18]. Another similar study was carried out by Vilar et al., where the PRP effect was evaluated in dogs with knee OA secondary to cranial cruciate ligament rupture. Lameness and gait improvements 3 months after IA injection were observed [161]. In horses with moderate to severe forelimb OA, PRP treatment did not cause statistically significant differences in gait, although the authors suggested a possible improvement in lameness despite the lack of statistical difference [162]. Another study in 16 rats with OA induced by monoiodoacetate injection showed promising results. Rats were randomly divided into treatment group A (*n* = 8 rats) or non-treatment group B (*n* = 8 rats). The treated group (A) received a single 0.5 mL injection of activated PRP 18 days after the monoiodoacetate injection, and histological assessment of treated joints revealed a higher chondrocyte population and greater cartilage thickness than in untreated animals [163]. Moreover, a study in rabbits in which the perichondral sheaths were dissected and the costal cartilages were removed concluded that the administration of PRP is helpful in improving chondrocyte density, resulting in increased ECM [164].

In humans, a review carried out by Fotouhi et al. concluded that PRP in combination with SVF showed an extreme reduction in pain along with a gradual increase in the cartilage thickness 3 and 6 months after treatment administration in patients with knee OA [165]. Another study evaluated the effect of PRP in chondrocytes apoptosis induced by IL-1β-, and concluded that PRP significantly decreases the chondrocyte apoptosis and modifies the apoptosis-associated expression at genes level [166].

Recently, Cugat et al. published two cases of full-thickness knee cartilage injuries treated with a novel autologous-made scaffold consisting of hyaline cartilage chips combined with a clot of a combination of plasma poor and rich in platelets and PRGF, respectively, with successful results [167]. This method has been recently named the CN-Biomatrix, and has demonstrated successful outcomes with good function in 15 patients with full-thickness cartilage defects [4]. Moreover, the same group performed a study in a sheep model where the same technique was successfully applied. Histological and immunohistochemical analysis demonstrated that this technique was appropriate to restore hyaline cartilage with adequate presence of type II collagen and minimum presence of type I collagen [168].

A combination of IA and intraosseous (IO) PRP infiltration has also been proposed with promising results. Better outcomes at 6 and 12 months were obtained in patients when PRP was applied both IO and IA, compared to the IA application alone [169,170].

The therapeutic potential of PRP products to treat OA is not fully understood. Further studies focused on the combination of other regenerative therapies with PRP are necessary, as well as more investigations in the field of IO infiltration.

**Table 1 biomedicines-09-01726-t001:** Research on osteoarthritis with MSCs and PRP-based therapy. Clinical studies.

Study	Model	Therapy	OA Location	Results
Hwang et al., 2021 [58]	Human	MSCs	Knee	MSCs therapy has shown potential in OA treatments with reduced pain, improved joint function, and enhanced overall life satisfaction in patients.
Chen et al., 2020 [132]	Human and Animals	PBMSCs	-	PBMSCs for cartilage repair achieved significant clinical improvement. Using PB as a source of seed cells enhances cartilage repair in vivo without serious adverse events
Cugat et al., 2020 [4]	Human	CN-Biomatrix + PRGF	Knee	Excellent clinical, functional, and MRI-based outcomes in young, active individuals with full-thickness cartilage or osteochondral defects.
Doyle et al., 2020 [103]	Human	BMMSCs	Knee	Improvements in pain, function, and life quality.
Jeyaraman et al., 2020 [49]	Human	ADSCs BMMSCs	Knee	At 6, 12 and 24 months, ADSCs showed significantly greater improvements than BMMSCs, compared to controls.
Freitag et al., 2019 [73]	Human	ADSCs	Knee	Both treatment groups receiving ADSCs showed clinically significant pain and functional improvement at 12-month follow-up. Radiological analysis using the MOAKS indicated modification of disease progression.
Hong et al., 2019 [91]	Human	SVF vs. HA.	Knee	No significant baseline differences were found between two groups. The SVF-treated knees showed significant improvement in the mean VAS, WOMAC scores, and ROM at 12-month follow-up compared with baseline. In contrast, the mean VAS, WOMAC scores, and ROM of the control group became even worse from baseline to the last follow-up visit. WORMS and MOCART measurements revealed a significant improvement in articular cartilage repair in SVF-treated knees compared to HA-treated knees.
Lee et al., 2019 [68]	Human	ADSCs	Knee	Single injection of ADSCs leads to a significant improvement in the WOMAC score at 6 months, while no significant changes were observed in control group. In MRI, there was no significant changes in cartilage defect at 6 months in the MSCs group whereas the defect in the control group was increased. An IA injection of autologous ADSCs provided satisfactory functional improvement and pain relief.
Rasheed et al., 2019 [154]	Human	PRP	Knee	Mean WOMAC score was significantly reduced at 6-month follow-up. Greater improvements were observed in the subgroup with patients having symptoms for less than 2 years.
Sánchez et al., 2019 [170]	Human	PRP	Knee	PRP IA injections in severe knee OA were not effective and did not provide any benefit. Combination of IA and IO infiltrations of PRP was not clinically superior at 2 months, but it showed superior clinical outcomes at 6 and 12 months.
Lamo-Espinosa et al., 2018 [113]	Human	BMMSCs with HA	Knee	BMMSCs-administered patients were superior according to WOMAC. MRI (WORMS protocol) showed that joint damage decreased only in the BMMSC high-dose group.
Song et al., 2018 [71]	Human	ADSCs	Knee	IA injections of ADSCs were safe and improved pain, function, and cartilage volume of the knee joint.
Spasovski et al., 2018 [62]	Human	ADSCs	Knee	Significant improvement in all four clinical scores (KSS, HSS-KS, T-L and VAS) was observed within the first 6 months, and improvement persisted throughout the rest of the follow-up. MOCART score showed significant cartilage restoration, whereas radiography showed neither improvement, nor further joint degeneration.
Huang et al., 2017 [151]	Human	PRP	Knee	The parameters VRS, functional score, and WOMAC Stiffness/Pain/Function score showed significant differences among the three groups. The three injections group had higher scores at 12-month follow-up.
Jo et al., 2017 [70]	Human	ADSCs	Knee	An IA injection of autologous ADSCs improved knee function measured with the WOMAC, KSS, and KOOS, and reduced knee pain measured with the VAS. Statistical significance was found mainly in the high-dose group. Clinical outcomes tended to deteriorate after 1 year in the low- and medium-dose groups, whereas those in the high-dose group plateaued until 2 years. The structural outcomes evaluated with MRI also showed similar trends.
Fodor et al., 2016 [87]	Human	SVF	Knee	At 3-months postoperative, there was a statistically significant improvement in WOMAC and VAS scores, which was maintained at 1 year. ROM and TUG both improved from preoperative to 3-months postoperative. Standard MRI assessment from preoperative to 3-months postoperative showed no detectable structural differences. All patients attained full activity with decreased knee pain.

MSCs: mesenchymal stem cells; OA: osteoarthritis; DHJD: degenerative hip joint disease; PBMSCs: peripheral blood mesenchymal stem cells; PB: peripheral blood; MRI: magnetic resonance imaging; ADSCs: adipose-derived stem cells; BMMSCs: bone marrow-derived mesenchymal stem cells; MOAKS: Magnetic Resonance Imaging Osteoarthritis Knee Score; SVF: stromal vascular fraction; VAS: visual analog scale; WOMAC: Western Ontario and McMaster Universities Osteoarthritis Index; ROM: ranges of motion; WORMS: Whole-Organ Magnetic Resonance Imaging Scores; MOCART: magnetic resonance observation of cartilage repair tissue; IA: intraarticular; PRP: platelet-rich plasma; KSS: Knee Society Clinical Rating System; HSS-KS: Hospital for Special Surgery Knee Score; T-L: Tegner–Lysholm; VRS: visual rating scale; KOOS: knee injury and osteoarthritis; TUG: timed up-and-go.

**Table 2 biomedicines-09-01726-t002:** Research on osteoarthritis with MSCs and PRP-based therapy. Preclinical studies.

Study	Model	Therapy	OA Location	Results
Okamoto-Okubo et al., 2021 [80]	Dog	PRP vs. ADSCs	Hip	Both therapies were apparently safe and effective in reducing chronic pain in dogs with bilateral DHJD during a 60-day period. However, a trend towards greater improvement was provided by the ADSCs treatment.
Tan et al., 2021 [142]	Rabbit Rat Mouse	MSCs-derived exosomes	−	This systematic review shows the therapeutic benefit of MSCs-derived exosomes therapy in cartilage repair.
Kriston-Pál et al., 2020 [76]	Dog	ADSCs	Elbow (42 animals), hip (5), knee (8), ankle (2), and hock (1).	83% of the OA patients improved or retained improvement in lameness at 4–5-year follow-up based on the owners’ subjective observations.
Murata et al., 2020 [84]	Rabbit	ADSCs	Knee	Total 2D-MOCART scores were higher in the implanted defects than in the controls, but not to a statistically significant extent. Similarly, average histological scores were comparable among all groups, although average gross scores were significantly higher in implanted defects than in controls.
Qiong et al., 2020 [127]	Rat	S-MSCs	Knee	In vivo study showed S-MSCs cell therapy significantly decreased serum inflammatory factor levels and protected cartilage by upregulating the expression of chondrogenic genes of meniscus chondrocytes derived from OA rats.
Takagi et al., 2020 [69]	Rabbit	ADSCs	Knee	Macroscopically, OA progression was significantly milder in the ADSCs sheets than in the control groups. Histologically, control knees showed obvious erosions in the medial and lateral condyles, while cartilage was retained predominantly in the ADSCs sheets group. Immunohistochemically, MMP-1, MMP-13 and ADAMTS-4 were less expressive in the ADSCs sheets than in the control groups.
Dominguez et al., 2019 [168]	Sheep	CN-Biomatrix + PRGF	Knee	Enhanced chondrogenesis and regenerated hyaline cartilage nearly normal under macroscopic ICRS assessment. Histological analysis showed equivalent structure to mature cartilage tissue in the defect and a collagen expression pattern in the newly formed cartilage similar to that found in adjacent healthy articular cartilage.
Olsen et al., 2019 [83]	Dog	IV Allogeneic MSCs	Elbow	A significant improvement in mean CSOM activity score and CSOM behavior score was observed when pre-treatment values were compared with post-treatment values (day >28). In contrast, mean PVF significantly decreased from baseline to post-treatment and weekly activity counts did not change. Synovial fluid biomarkers did not change during treatment, and labeled MSCs were rarely detected in synovial fluid samples collected after MSC administration.
Wu et al., 2019 [122]	Porcine	hUC-MSCs with HA hydrogel transplanted	Knee	The treated knees showed significant gross and histological improvements in hyaline cartilage regeneration when compared to the control knees. The ICRS histological score was higher for the treated knees than the control knees.
Feng et al., 2018 [74]	Sheep	ADSCs with HA	Knee	Evaluations by MRI, macroscopy, micro-computed tomography, and cartilage-specific staining demonstrated that the ADSCs + HA-treated groups preserved typical articular cartilage features. Inflammatory factors from synovial fluid of ADSCs + HA treated groups were significantly lower than those in the HA alone group.
Vilar et al., 2018 [161]	Dog	PRP	Knee	Dogs with CCLR treated with IA PRP had improved PVF, VI, ST, and ROM over time.
Sengul et al., 2017 [164]	Rabbit	PRP	Knee	In the PRP and sham groups, the volumes of the cartilages and perichondrial sheaths were higher than those of the control group. The numerical densities of the chondroblasts and chondrocytes increased more in the PRP group than in the sham group.
Wang et al., 2017 [139]	Mice	MSCs-derived exosomes	Knee	IA injection of MSCs-derived exosomes alleviated cartilage destruction and matrix degradation in the DMM model.
Yun et al., 2016 [66]	Dog	PRP and ADSCs	Knee	The lameness scores were lower, and the focal compression strengths of the affected femoral articular surface cartilages were higher for treated dogs than for those in the OA control group. Additionally, the inflammatory changes, when evaluated with Mankin scoring and histomorphologic examination, were significantly ameliorated with PRP and/or ADSCs treatment.

DHJD: degenerative hip joint disease; ADSCs: adipose-derived stem cells; MSCs: mesenchymal stem cells; OA: osteoarthritis; MOCART: magnetic resonance observation of cartilage repair tissue; S-MSCs: synovial mesenchymal stem cells; MMP-1: matrix metalloproteinase1; MMP-13: matrix metalloproteinase 13; ADAMTS-4: disintegrin and metalloproteinase with thrombospondin motifs 4; ICRS: International Cartilage Repair Society; CSOM: client-specific outcome measure; PVF: peak vertical force; MRI: magnetic resonance imaging; HA: hyaluronic acid; CCLR: cranial cruciate ligament rupture; IA: intraarticular; PRP: platelet-rich plasma; VI: vertical impulse; ST: stance time; ROM: ranges of motion; DMM: destabilization of the medial meniscus.

### 3.4. Future Perspectives in OA

In OA, bioregenerative therapies have been demonstrated to be a great option over other treatments due to their therapeutic potential. In recent years, OA cell therapies have been developed as an alternative or additional therapy to traditional methods, with the aim of creating a new tissue displaying the most similar characteristics to native cartilage.

The optimal OA therapy should halt disease progression through the repopulation of the injured tissue with chondrocytes able to produce a hyaline matrix, restoring cartilage structural and functional properties. Different new approaches for cartilage regeneration have been proposed, for example, anti TNF-α therapies, as TNF-α plays a considerable role in OA pathogenesis [171]. Moreover, it has been demonstrated that other biologic agents that inhibit nerve growth factor improve function and reduce pain in OA patients [172]. Future perspectives are focused on gene therapy with encoding genes for chondrogenic GFs and anti-inflammatory cytokines; particularly, this therapy exerts its effects through intracellular nucleic acid transfection and translation into protein [173]. Recently, therapeutic strategies that combine cell and gene therapy based in the production of protein platforms have been proposed as good strategies to treat OA [174,175]. The main limitation of gene therapy is its carcinogenic potential as well as the elevated costs [176].

Other important considerations include the time that the therapeutics stay in the joint tissue. Different natural and synthetic scaffolds (amphiphilic polymeric micelles and hydrogels) are being evaluated to achieve increased articular dwelling of the drugs [177]. Moreover, several studies have reported the beneficial results of nanoparticles used for targeted drug delivery and sustained release in OA joints [178].

Nanotechnological strategies combined with cell-based therapy, biological and gene treatments could be a future perspective in the clinical management of the pathology.

### 3.5. Study Limitations

Numerous studies regarding cell therapy to treat OA are currently being conducted due to their safety and their ability to relieve pain and improve function. That said, the authors consider that the reviewed studies have several limitations, including small sample sizes, short-term follow up periods, absence of control groups, and inconsistent methodology. Thus, further studies taking into account these considerations are required.

To overcome these limitations, further studies with a greater sample size are needed. Additionally, in our experience, it is necessary that all studies follow the same inclusion and exclusion criteria, and of them should consider the same outcomes and clinical data. Moreover, obtention protocols should be standardized, and security and quality must be verified. Furthermore, the approval of these products by a competent authority is mandatory.

## 4. Other Clinical Applications

### 4.1. Adipose-Derived Mesenchymal Stem Cells and Stromal Vascular Fraction

ADSCs, as well as SVF, can maintain self-renewal and enhanced multidifferentiation potential, allowing them to repair a large variety of damaged organs and tissues. Consequently, their application in tissue engineering and organ regeneration has been under study during the last few decades.

#### 4.1.1. Musculoskeletal Applications

ADSCs and SVF have been widely used for reconstructive surgery to repair segmental bone defects, critical-size defects and bone fractures, combined with appropriate scaffolds and bioactive factors [179,180]. ADSCs and SVF have been adopted as alternative therapies to treat calvarial defects, delayed fracture healing and avascular necrosis of femoral head, and have demonstrated promising results, such as new bone formation and increased angiogenesis of the new formed tissue compared to conventional treatment groups, as well as decreased pain and general discomfort reported by patients [181,182,183,184].

Moreover, the application of ADSCs and SVF to treat tendon injuries, such as Achilles tendinopathy, lateral epicondylosis and rotator cuff disease, has been studied. The results demonstrate that ADSCs therapy is effective in improving pain, performance and structural defects evaluated with MRI or ultrasound [185,186].

#### 4.1.2. Myocardium Regeneration

ADSCs therapy has been widely investigated as a prospective treatment for myocardial infarction in preclinical and clinical trials. Up to now, four different transplant methods have been investigated: intramyocardial injection, intravenous injection, intracoronary injection and cell spray transplantation [180]. All clinical trials showed similar results, including improved cardiac function and perfusion defects, reductions (up to 50%) in myocardial scar formation, marked increments in Minnesota Living with Heart Failure Questionnaire results, improved tolerance to exercise and a trend towards better Left Ventricular Ejection Fraction. Moreover, no adverse events have been reported during the procedure [187,188,189,190,191,192].

#### 4.1.3. Plastic and Reconstructive Surgery

The use of ADSCs and SVF therapies for the treatment of severe burn injuries and intractable ulcers has gained the interest of researchers and clinicians during the past few years, as encouraging results have been obtained in animal models. That said, only a few clinical trials have been performed to evaluate ADSCs treatments [180]. ADSCs were effective and safe when used for the treatment of radiation injuries, as well as for the treatment of skin facial defects [193,194]. The results show that the regenerated tissue quality was significantly superior to pre-treatment; rapid coverage of the wound with regenerated tissue and no recurrence or new ulcerations after treatment were reported. However, autologous fat grafts were reported to be insufficient to ameliorate mature pediatric burn scars [195]. The role of SVF in the treatment of scars in the face has also been studied, and patients who received SVF-enriched fat grafts showed better soft tissue volume maintenance compared to the control group [196]. Furthermore, it has been reported that chronic skin wounds treated with SVF healed better than those treated with hyaluronic acid [197].

Some articles reporting the clinical outcomes of autologous fat grafts enriched with SVF in breast reconstruction and breast augmentation have been published in the last few years. The results showed a restoration of breast contour and an increase in the three-dimensional volume, so its use seems to be safe and effective [198,199].

### 4.2. Platelet-Rich Plasma

The ubiquitous nature of the mechanism of action of PRP suggests that it can be used to treat several pathologies to aid the body’s natural healing processes [200].

#### 4.2.1. Musculoskeletal Applications

With regard to musculoskeletal pathologies, PRP has been applied in several conditions, such as tendinopathies, cartilage pathologies and acute muscle injuries [200].

Regarding tendinopathies, most of the research into PRP treatment focuses on lateral epicondylitis. A meta-analysis conducted by Ben-Nafa et al. concluded that PRP demonstrated better but delayed therapeutic effects compared to corticosteroid injections, with no subsequent regression of their positive clinical effects for up to 2 years, and with no reported adverse effects except for local discomfort [201]. PRP has also been used to treat patella tendinosis with good results. Dupley et al. found in their meta-analysis that PRP injections are statistically superior to the control treatment (dry needling and extracorporeal shockwave therapy) at six-month follow-up [202]. PRP injections have also been tested for other tendinous pathologies, such as Achilles tendinopaty [203] and rotator cuff disease [204], but no clinical benefit was observed. PRP has also been used as an augmentation for anterior cruciate ligament surgery, and there is evidence to suggest that adding PRP to the grafts could be beneficial in expediting graft maturity [205].

With regard to acute muscle injuries treatment with PRP, a systematic review conducted by Grassi et al. concluded that the return to sport time was significantly shorter in the PRP groups than in the control group; however, re-injury rates were very similar between PRP and controls, and there were no significant differences regarding muscle strength, flexibility, and muscle healing on ultrasound scan or magnetic resonance imaging [206].

#### 4.2.2. Plastic Surgery and Reconstructive Medicine

During the last decade, PRP has been widely use in plastic and reconstructive surgical applications. The efficacy of PRP in promoting wound healing, including chronic diabetic and nondiabetic ulcers, as well as acute traumatic wounds, has been deeply studied, and significantly favorable outcomes, such us better re-epithelization rate and shorter time to healing, have been reported [207,208,209].

Moreover, PRP has been combined with autologous fat grafts, particularly in facial plastic surgery, and superior contour restoration and volume maintenance, as well as improvements in facial scarring, have been described when compared with fat grafting alone [196,210]. Similar results are obtained when PRP is combined with breast fat graft, and improvements in breast volume maintenance have been observed [211].

In the field of plastic and reconstructive surgery, numerous researchers have investigated the effects of PRP on bone grafts survival rates, especially in maxillofacial surgery, and in general lines, these studies have shown that PRP-enriched bone grafts involve a shorter time to bone regeneration, decreased postoperative pain, and higher bone augmentation [209,212,213].

Among all these applications, PRP has also been used in aesthetic applications, such as skin rejuvenation procedures [214,215,216] and hair restoration, showing better outcomes than conventional treatments [216,217].

#### 4.2.3. Oro-Facial Tissue

The regeneration potential of PRP in oral tissues has revolutionized clinical dentistry in last years. PPR has been successfully used in the regeneration of periodontal tissues [218,219], especially in the treatment of intrabony defects, and has shown antibacterial properties for periodontal pathogens [220]. Additionally, the application of PRP after tooth extraction improves soft tissue healing and enhances bone regeneration [221]. Moreover, PRP placement around dental implants improves bone formation and implant osseointegration, thus longer survival rates are achieved [218,222,223], as well as alveolar bone regeneration [224].

Although good results have been obtained, more clinical studies are required in order to define the clinical relevance of the outcomes related to PRP in oral tissues.

#### 4.2.4. Ophthalmology

PRP has emerged as an alternative treatment for some ocular surface diseases, such as dry eye and corneal ulceration, with significant therapeutic potential [225].

Regarding primary and secondary dry eye diseases, several studies have reported that PRP eye drops significantly and immediately improve dry eye symptoms, including conjunctival hyperemia, corneal erosion, sensitivity to light, and eye burning sensation, with low risk of adverse effects [226,227,228,229].

On the other hand, autologous PRP eye drops have been reported in numerous studies as an effective treatment for corneal ulcers. A reduction in depth and size, or complete closure of the ulcer, is experienced by most of the patients after treatment, and the improvement of associates symptoms, such as inflammation and pain, as well as improvements in visual acuity have been reported by patients [230,231,232]. PRP clots have also been used in the treatment of perforated corneal ulcers, and treated eyes showed reduced inflammation and pain, with most of them healing completely [233,234]. Moreover, PRP has been shown to be effective in the treatment of corneal burns. Reduced corneal and conjunctival healing time was observed, together with improved corneal transparency and visual acuity, with PRP compared to conventional topical treatment [235,236].

## 5. Conclusions

Cell and cell-free therapies could be a non-invasive and more effective treatment alternative, which are less invasive than surgery and conventional therapies.

Cellular therapy is a current modality validated to treat OA, and involves the combination of conventional treatments with the application of cell-based therapies. It is very important to provide patients with alternative or minimally invasive therapy options that may significantly halt disease progression.

We should consider that it has been evidenced that cultured MSCs can undergo a malignant transformation, with the ability to develop malignancy in immunodeficient patients. However, in the last few years, MSCs have been shown to be immunoprivileged, genetically stable in long-term cultures and with low risk of rejection. With regard to PRP, a standardized obtention protocol is needed, and we should emphasize that pain and inflammation at the injection site have been reported as frequent side effects.

To conclude, in this review, a novel and promising treatment approach for patients with degenerative OA that is safe, one-step, cost-effective, minimally invasive, and can be conducted with autologous cells, has been reviewed. However, while studies show that regenerative therapies have a positive effect on OA patients, more clinical trials with high-quality evidence and long-term follow-ups are needed.

## Figures and Tables

**Figure 1 biomedicines-09-01726-f001:**
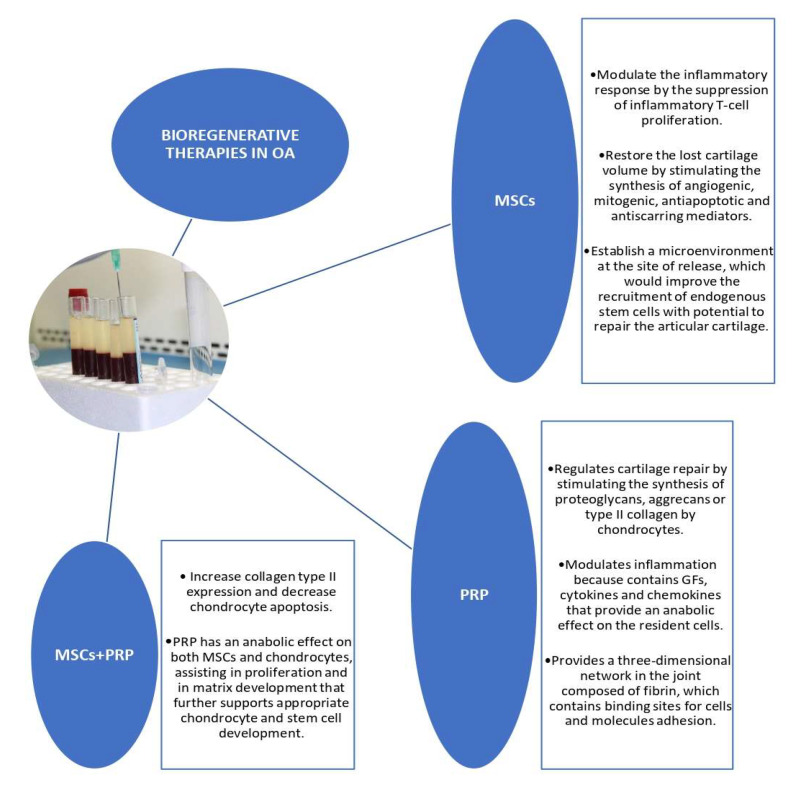
Graphical abstract showing the role of MSCs and PRP in OA. OA: osteoarthritis; MSCs: mesenchymal stem cells; PRP: platelet-rich plasma; GFs: growth factors.

## Data Availability

Not applicable.

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
