# Peer review of "Cell and Cell Free Therapies in Osteoarthritis"

_biomedicines, 2021, doi:10.3390/biomedicines9111726_

Round 1

Reviewer 1 Report

Summary

This review article entitled “Cell Therapy in Osteoarthritis” focused on cell and cell free based therapies as promising alternative for osteoarthritis management against the available conventional medical or surgical treatments. In this article authors mainly focused on mesenchymal stem cells or medical signaling cells (MSCs) and platelet rich plasma (PRP) as cell and cell free therapies. Further author’s elaborated on various Mesenchymal Stem Cell sources that can be of more appropriate use for Osteoarthritis including adipose tissues, stromal vascular fraction, bone marrow and other sources including umbilical cord, synovium, and peripheral blood. Further authors described cell-free therapies including extracellular vesicles and Platelet Rich Plasma.

Overall authors present an interesting review. However, they should address my comments below before considering for publication.  

 Comments

  1. Authors should consider incorporating “cell free” in the title as they discussed on extracellular vesicles and platelet rich plasma (PRP) in this article.
  2. Materials and Methods should be replaced with literature search methods.
  3. Extracellular vesicles and PRP are free of cells, authors should emphasis this point in the respective sections.
  4. Consider providing separate tables for preclinical studies and clinical trial studies.
  5. Authors should advocate the reads, how to overcome the limitations based on their best experiences.  
  6. Page #4, line 145, “PRP has been used as an OA therapeutic option” this sentence should be supported by suitable reference.
  7. Consider to provide a separate section “cell free therapies for OA” for “Mesenchymal Stem Cells Exosomes in Osteoarthritis and Platelet Rich Plasma in Osteoarthritis”.  

Reviewer 2 Report

Dear Authors, I have read the manuscript, I think that it is complete and very well written, only minor points:

  • Introduction: line 53 reference 12: is:"Damia, E.; Chicharro, D.; Lopez, S.; Cuervo, B.; Rubio, M.; Sopena, J.J.; Vilar, J.M.; Carrillo, J.M. Adipose-Derived Mesenchymal 737 Stem Cells: Are They a Good Therapeutic Strategy for Osteoarthritis? Int J Mol Sci 2018, 19, 1926. This reference must be deleted and must be changed to these:  doi: 10.1007/s00296-019-04332-6; doi: 10.1016/j.joca.2013.06.026; doi: 10.2165/00044011-200727020-00004.

Conclusion: Please add the risk, of these methods. 

Round 2

Reviewer 1 Report

None